# Prevention and Treatment Strategies for Respiratory Syncytial Virus (RSV)

**DOI:** 10.3390/pathogens12020154

**Published:** 2023-01-17

**Authors:** Dvir Gatt, Isaac Martin, Rawan AlFouzan, Theo J. Moraes

**Affiliations:** 1Division of Respiratory Medicine, Department of Pediatrics, Hospital for Sick Children, Toronto, ON M5G 1X8, Canada; 2Program in Translational Medicine, Hospital for Sick Children, Toronto, ON M5G 1X8, Canada

**Keywords:** respiratory syncytial virus, antiviral, vaccine, bronchiolitis, monoclonal antibodies

## Abstract

Respiratory syncytial virus (RSV) is a leading cause of severe lower respiratory tract disease, especially in young children. Despite its global impact on healthcare, related to its high prevalence and its association with significant morbidity, the current therapy is still mostly supportive. Moreover, while more than 50 years have passed since the first trial of an RSV vaccine (which unfortunately caused enhanced RSV disease), no vaccine has been approved for RSV prevention. In the last two decades, our understanding of the pathogenesis and immunopathology of RSV have continued to evolve, leading to significant advancements in RSV prevention strategies. These include both the development of new potential vaccines and the successful implementation of passive immunization, which, together, will provide coverage from infancy to old age. In this review, we provide an update of the current treatment options for acute disease (RSV-specific and -non-specific) and different therapeutic approaches focusing on RSV prevention.

## 1. Introduction

In light of the current COVID-19 pandemic, there is increasing awareness of the morbidity and mortality associated with respiratory viruses. It is important to note that respiratory viruses were significant health concerns long before this pandemic. Specifically, respiratory syncytial virus (RSV) is the leading global cause of acute lower respiratory infections in children under 2 years of age [1]. 

While almost all infants will be infected by the age of 2, a subset will have significant illness requiring hospital admission. In Canada, approximately 2% of all infants are admitted to hospital with an RSV infection in the first 2 years of life. While high-risk infants in developed countries have access to monoclonal antibodies designed to reduce illness severity, the overall hospital admission rates have not changed in North America, as most admitted infants are not considered high-risk [2,3]. RSV admissions represent a significant health care burden, and in Ontario alone (the population was ~13 million at the time of the study), the estimated costs attributed to *hospitalized* RSV patients averaged over 10 years was approximately CDN $13,000,000 per year [4].

Mortality in developed countries is not common, occurring in less than 10 individuals per 1,000,000 live births [4]. However, globally, RSV is a leading cause of death, and worldwide, one in every 50 deaths among children between 0 and 5 years and one in every 28 deaths among children aged 1 month to 6 months are attributable to RSV [5]. It is estimated that of the 101,400 global deaths secondary to RSV in 2019, 97% occurred in low- and middle-income countries [5].

In addition to acute disease, there is evidence suggesting that RSV infection in childhood may trigger persistent or recurrent wheezing and asthma in later life, linking RSV morbidity to chronic illness [6,7,8]. Despite the disease burden associated with RSV, there is a paucity of specific and non-specific approaches used to treat or prevent infection. However, novel agents are on the horizon and offer hope as a means to mitigate the impact of this virus. Here, we review the current and near-future state of the clinically available RSV therapies and prophylaxis agents. Specifically, we discuss agents that are specifically for RSV treatment, agents or approaches that are non-specific but still have been considered for RSV treatment, and, finally, approaches designed to prevent RSV, including the most recent vaccine developments.

## 2. Specific Respiratory Syncytial Virus Therapy

RSV is a linear single-stranded RNA virus that encodes 11 proteins. The virus has two transmembrane glycoproteins that are involved in viral entry into the cells, which are attachment (G) and fusion (F) glycoproteins. Glycoproteins are involved in binding to the cell surface, whereas F glycoproteins facilitate fusion to the cell membranes. The virus is mainly transmitted by direct contact and, to lesser extent, via the droplet route, as the virus can only be aerosolized by large droplets. RSV has an incubation period of 2–8 days and initially results in an upper respiratory tract infection. Subsequently, a lower respiratory tract infection can develop secondary to aspiration or intracellular transmission. RSV infection is typically a self-liming disease in otherwise healthy children. Thus, supportive treatment remains the mainstay of therapy. However, RSV-targeted therapy has demonstrated benefits in selected pediatric populations, such as bone marrow and lung transplant recipients. The ultimate goals of these therapeutic measures are to alleviate symptoms, decrease the duration and severity of the illness, and decrease the risk of transmission. Treatment options that have been studied include ribavirin, palivizumab, motavizumab, and RSV-immune globulin (RSV-IVIG). 

**Ribavirin** is a broad-spectrum nucleoside analog that inhibits the replication of DNA and RNA viruses. It is available in aerosolized, oral, and intravenous (IV) formulations. An older Cochrane review included 12 randomized trials comparing ribavirin with placebo in infants and children with RSV-positive lower respiratory tract infection. In four trials, the difference in mortality was not statistically significant (OR 0.37; 95% CI, 0.12 to 1.18). In addition, there were no statistically significant differences in the outcomes between the treatment group and placebo, including the length of hospitalization, illness severity, and improvement in oxygenation [9]. The aerosolized formulation requires specialized inhalation devices, such as the small-particle aerosol generator model-2 (SPAG-2). Aerosolized ribavirin should only be administered in a well-ventilated room with at least six air exchanges per hour in order to minimize the possibility of inadvertent exposure to others in the patient’s room. Patients should ideally be allocated to a negative pressure room, and the SPAG-2 device should be turned off for 5 min prior to lifting any hood or tent used [10]. 

A randomized control trial (RCT) was carried out among bone marrow transplant recipients who tested positive for RSV [11]. Eligible subjects were randomized to receive aerosolized ribavirin combined with supportive care or supportive care alone. A total of 2 g of ribavirin inhalation solution at a concentration of 60 mg/mL was administered over 2 h 3 times daily for a total of 10 days. The study demonstrated that ribavirin treatment was associated with a reduction in the viral load and pneumonia.

The reported side effects included bronchospasm, shortness of breath, chest pain, skin rash, and conjunctivitis. Other side effects included headache and vomiting, which could also affect healthcare providers and family members, despite applying the proper method of delivery. In order to evaluate airborne ribavirin exposure among health care providers, its post-work shift urinary concentration was measured. The medication was detected in 62% and 12% of the urinary samples provided by the nurses and respiratory therapist, respectively. Ribavirin has demonstrated a teratogenic effect on animals when studied in rabbits and rats at a dose of 1 mg/kg. However, these teratogenic effects were not observed at a lower dose of 0.3 mg/kg, which is equivalent to 0.05 mg/kg when adjusted for the adult body surface area in humans. Although teratogenic effects were not observed in the offspring of exposed health care workers, aerosolized ribavirin exposure remains a concern [12,13].

Oral ribavirin for lung transplant recipients with RSV infection was previously reviewed [14]. A loading dose of IV ribavirin (33 mg/kg in three divided doses) was administered to 52 lung transplant recipient patients, while 2 other patients received an equivalent dose orally. Twenty-one patients were then given oral ribavirin (20 mg/kg in two divided doses) for 6–31 days, with a median duration of 11 days. Oral ribavirin appears to be an effective alternative to IV ribavirin in the treatment of RSV infection after lung transplant. Moreover, a meta-analysis and systemic review showed a statistically significant reduction in mortality among patients with hematological diseases who were treated with ribavirin [15]. Oral ribavirin was associated with good viral clearance, rendering it a safe, easy, and cost-effective alternative compared the aerosolized form. 

However, overall, due to the high cost and potential adverse side effects, ribavirin is not currently recommended for previously healthy children. Given that benefits have been observed in severely immunocompromised patients, the treatment’s clinical use among these populations remains a consideration.

Other promising antiviral agents are currently being investigated in randomized control trials, such as RV521 (NCT04225897) [16] and AK0529 (ziresovir) (NCT04231968) [17], amongst other compounds, with both agents employing the use of fusion or replication inhibition as their mechanism of action.

**Palivizumab** is a recombinant humanized monoclonal immunoglobulin that was approved in 1998 for prophylaxis against severe RSV disease in select high-risk groups. Palivizumab is effective in binding to the RSV fusion protein and inhibiting subsequent viral infection. Given its efficacy in prevention, studies have examined the possibility that palivizumab may be used to treat acute infection. In a double-blinded RCT, 420 previously healthy infants with RSV bronchiolitis were randomly assigned to palivizumab (15 mg/kg) or placebo [18]. The palivizumab group did not demonstrate a significant difference from the placebo group in the hospital readmission rate, hospitalization stay duration, or pediatric intensive care unit PICU admission. Another multicenter RCT that randomly assigned proven RSV-positive acute bronchiolitis patients to palivizumab or placebo did not reveal statistically significant differences in clinical outcomes between the groups [19].

**Motavizumab** is a second generation humanized anti-RSV monoclonal antibody developed from palivizumab. As in the case of palivizumab, investigators have explored whether motavizumab offers any therapeutic benefit. A multicenter RCT, the goal of which was to study the effect of motavizumab on RSV-positive pediatric patients [20], randomly assigned subjects to three arms: motavizumab 30 mg/kg, motavizumab 100 mg/kg, and placebo groups. There was no difference in the duration of hospitalization, severity of illness, or future wheezing episodes in the children treated with motavizumab or placebo during the 12-month follow-up.

In addition to antiviral and monoclonal antibodies, **RSV-IVIG** is another treatment option that has been studied for acute RSV infection. This intravenous polyclonal immunoglobulin has a strong neutralizing effect on RSV (in animal models) and was produced by isolating RSV antibody from pooled blood. Trials have shown a reduction in RSV replication in the lung, thereby reducing the viral load and preventing the subsequent development of respiratory illness in animal models [21]. One double-blinded RCT evaluated the role of RSV-IGIV in treating RSV infection in a “high-risk” population: 107 children less than 2 years of age with prematurity, congenital heart disease, or chronic lung disease. Patients in the active treatment arm were infused once intravenously with a relatively large volume of the drug at a dose of 1500 mg/kg or placebo. There were no differences in the duration of hospitalization or the severity of illness in either arm [19]. RSV-IGIV was administered in a relatively large volume intravenously, since it is associated with the potential risk of blood-born product transmission and possible interference with the administration of routine live vaccines. A 2019 Cochrane review examines seven trials involving 486 infants with RSV bronchiolitis and concluded that there was insufficient evidence of a difference between immunoglobulins and placebo to determine any review outcomes [22]. In 2003, RSV-IGIV was withdrawn voluntarily from the market when other suitable alternatives became available [23].

## 3. Non-Specific RSV Treatments

### The Lung in RSV Infection

In the exploration of non-specific therapies that have been proposed and trialed for RSV bronchiolitis/infection, it is important to understand the pathophysiology of an acute infection. RSV infects and replicates in the mucosa lining the respiratory tract from the nasopharynx to the distal alveoli. The effects of RSV infection in adults and older children tend to be related to upper respiratory tract pathologies, including rhinorrhea and coryza, symptoms of the common cold. In children, particularly infants and preschool children, viral effects on the lower respiratory tract are more predominant and lead to the more severe and potentially life-threatening sequelae of this infection, including bronchiolitis. 

There are several proposed reasons for this difference. The first has to do with scale, as the average diameter of the adult respiratory bronchiole is approximately 250 µm, compared to only 120 µm in an infant of 4 months [24]. This vastly smaller bronchiolar lumen is more easily occluded during viral infection [25]. Additionally, the sources of alveolar collateral ventilation that are present in older children and adults (the pores of Kohn and canals of Lambert) are relatively immature, rendering the infant lung more prone to obstruction [26]. These physical properties of the lung, when combined with a developing immune system, render the infant more susceptible to clinically meaningful episodes of bronchiolitis. 

Further insights into the mechanism of RSV bronchiolitis have been obtained through the examination of post-mortem lung samples of infants who died from severe RSV infection. Histological analyses of such patients have shown that most pathological changes in acute bronchiolitis involve the medium and small bronchioles (150 μm and smaller). In these infected bronchioles, airway obstruction is caused by airway edema, epithelial cell injury with the accumulation of inflammatory cells and other cellular debris, and increased airway mucus [26,27]. This obstruction is associated with increased airway resistance and leads to hyperinflation, hypoxemia, increased work in breathing, crackles, and wheeze. 

Normal mucus comprises the fluid lining that protects the airway and traps particulate matter inhaled from the environment. The synchronized movement of the cilia, themselves bathed in a periciliary layer of fluid, works to transfer the mucus to the upper respiratory tract. This mucociliary escalator, when combined with an effective cough, is the body’s first line of defense against environmental pathogens and debris. In bronchiolitis, this mechanism is impaired, partly due to increased mucus production and partly due to viral effects on the cilia. Several groups have examined infants with RSV infection who did not have significant comorbidities (e.g., prematurity, bronchopulmonary dysplasia, congenital heart disease) and identified genes associated with inflammatory pathways implicated in severe disease [28]. This complex interplay between the viral genome and host immune system can result in a type-2 immune response and the release of pro-inflammatory cytokines that, in turn, activate mucus metaplasia and mucin secretion, further exaggerating airway obstruction [29,30,31]. An understanding of these potential mechanisms has led to various therapeutic approaches that can be applied to reverse airway obstruction, most notably mucolytics [32], bronchodilators [33], and anti-inflammatory agents [34], and they are summarized in Table 1. Interestingly, despite the mechanistic links, the evidence does not support the notion that these approaches are of universal benefit for RSV infections.

**Dornase alfa**: The mucus plugs observed in RSV bronchiolitis contain large amounts of extracellular DNA, a by-product of leukocyte recruitment and subsequent degradation [36]. Human recombinant DNase (hrDNase) is a compound that cleaves this extracellular DNA and reduces the viscosity of mucus and has been proposed as a treatment for bronchiolitis due to these properties. Although smaller studies and case reports have shown improvements in chest radiograph atelectasis for bronchiolitis in infants with severe disease [50,51], the largest randomized, double-blinded, controlled trial involved 225 hospitalized infants with bronchiolitis and failed to demonstrate any significant difference in the length of hospital stay or the duration of supplemental oxygen administration [52]. A Cochrane review conducted in 2012 included this trial and two others, totaling 333 children aged up to 24 months, and found no difference when the treatment was used for infants hospitalized with viral bronchiolitis [35]. Thus, nebulized hrDNase can be considered in severe cases of atelectasis associated with bronchiolitis when conventional treatment options have failed but is not generally recommended as a treatment for RSV infection. 

**N-acetylcysteine (NAC)** is a compound that hydrolyzes the disulfide bonds of mucins and has additional antioxidant effects, both properties that may be of use in bronchiolitis [53]. In an in vitro infection model of alveolar type-II epithelial cells infected with RSV, NAC was shown to inhibit mucin synthesis and reduce the production of pro-inflammatory mediators [54]. However, there are no high-quality, randomized, placebo-controlled trials investigating its use for bronchiolitis. A search of the literature revealed only one randomized, controlled study of 100 infants, which compared nebulized NAC with nebulized salbutamol and was inadequately powered to demonstrate improvements in the symptom score or duration of hospital admission [55]. Further studies are therefore required before NAC can be considered as a therapy for RSV infection. 

**Nebulized hypertonic saline** increases mucociliary clearance in healthy subjects, as well as those with asthma [56], bronchiectasis [57], and cystic fibrosis [58]. It is understood to work by creating an osmotic gradient, pulling water into the mucus layer from the mucosa and submucosa, thereby also potentially reducing airway edema [39,59]. The hydration of the mucus layer helps to mobilize secretions, and when combined with the stimulation of an effective cough, it should alleviate airway obstruction [39,60]. Despite these mechanistic paradigms, the administration of 3% hypertonic saline was not shown to improve wheezing or the airflow when assessed both clinically and with computerized acoustic airflow techniques [61]. The results of clinical trials investigating the efficacy of hypertonic saline in viral bronchiolitis are also disappointing. Most clinical trials demonstrated, at best, a modest effect of 3% hypertonic saline when the length of hospital stay and symptom scores were examined [60,62]. A 2017 Cochrane meta-analysis of 28 trials involving 4195 infants with acute bronchiolitis concluded that there was insufficient evidence to recommend the use of hypertonic saline, though the authors did note that the results seemed to favor a modest reduction in the length of stay, with very few minor and adverse events [40]. Subsequent randomized clinical trials have also failed to demonstrate any benefits of 3% hypertonic saline when compared to standard supportive care [63,64]. Therefore, it is still not a recommended standard treatment for RSV bronchiolitis. 

**Bronchodilators**: When auscultating an infant with bronchiolitis, wheeze may be audible, and a prolonged expiratory phase with crackles is characteristic. Salbutamol and albuterol are two β-2 adrenergic agonists which act to relax the airway smooth muscle and relieve obstruction. While early meta-analyses suggested that use of these agents was associated with a moderate short-term improvement in some mild or moderate cases [65], a 2014 Cochrane meta-analysis of 30 studies comprising 1992 infants with bronchiolitis demonstrated no significant differences in oxygen saturation, the rates of hospital admission, or the duration of hospitalization [33]. Furthermore, the administration of β-2 adrenergic agonists, such as salbutamol or albuterol, is associated with a side effect burden including tachycardia, oxygen desaturation, tremors, and electrolyte abnormalities. Therefore, bronchodilators are not recommended routinely as a therapeutic option for bronchiolitis but can be trialed to assess the response in infants when wheeze is the predominant feature used to assess the response. Although data for this are lacking, the heterogeneity of the presentations and symptoms of RSV is widely recognized, and the selected use of bronchodilators for certain subgroups of patients based on the clinical phenotype could be the starting point for future targeted, randomized clinical trials [66]. 

**Epinephrine** has a theoretical effect on acute bronchiolitis because of its secondary effects on the β-2 adrenergic receptors, leading to the relaxation of the airway smooth muscle, as well as its strong alpha-adrenergic properties, leading to vasoconstriction and a reduction in airway edema [42]. In one small study, racemic epinephrine was found to be superior to salbutamol in improving both airway resistance and clinical scores among infants with bronchiolitis [67]. A 2011 Cochrane Review included 19 studies, of which 9 (1354 infants) compared nebulized epinephrine to placebo, demonstrating a modest short-term improvement in the outpatients. However, this had no effect on illness progression when evaluated at day 7 and was not associated with shorter hospital stays or improved symptom scores among the inpatients [42]. Notably, the tolerability was good in the studies examined here, with very few adverse events. Thus, although epinephrine may be used in the acute setting, it is not recommended as a standard treatment for the management of RSV infection. 

**Glucocorticoids**: Several studies have shown that glucocorticoids have limited anti-inflammatory properties in the context of RSV infection with respect to both viral load and cytokine production [68,69]. However, clinical studies have shown no beneficial effect in reducing the clinical scores, hospitalization rates, or length of hospital stay for steroid use. One potential reason for this is that neutrophilic inflammation predominates in RSV infection [70], and this type of inflammation is known to be poorly responsive to glucocorticoids [69]. A 2013 Cochrane review of 17 controlled studies involving 2596 infants with bronchiolitis demonstrated that steroid therapy does not affect the clinical course of infants and children admitted to hospital with bronchiolitis, and their use is not recommended for the management of bronchiolitis in otherwise healthy, unventilated patients [45]. Certain groups have also hypothesized that inhaled glucocorticoids might confer a lower risk of later developing asthma in RSV-infected infants, but the results of a 6-year follow-up study of 185 infants in the Netherlands did not support this hypothesis [44]. There are, however, potential instances where infants with RSV may benefit from steroids, such as those with underlying bronchopulmonary dysplasia and asthma, perhaps because these subgroups may have a greater degree of steroid-responsive inflammation than other children with RSV infection. Although there is no good-quality data to support this approach, one trial of 200 infants admitted with bronchiolitis found that dexamethasone reduced the length of stay among those with eczema or a family history of asthma in a first-degree relative [71]. Further trials are required before this can become a standardized approach. 

**Leukotriene receptor antagonists**: RSV-triggered bronchial hyperresponsiveness and mucus hypersecretion are understood to be partly due to the production of leukotrienes, which induce bronchoconstriction via interaction with the bronchial smooth muscle [46]. A Cochrane review of five randomized, placebo-controlled studies with a total of 1296 infants aged <24 months who were hospitalized with bronchiolitis examined the data associated with leukotriene receptor antagonists and found that the quality of evidence was so poor that it did not allow for any conclusions on the reductions in the length of hospital stay or clinical severity score [47]. Pending further trials, this class of therapeutic agents for RSV infection cannot be advocated. 

**Other Therapies:** There have been many other proposed non-specific therapies for acute RSV infection that have been investigated, including antibiotics, such as azithromycin [72], and various combinations of the agents described above [67,73]. The combined use of nebulized epinephrine with 3% hypertonic saline, for example, has shown promising results in improving the clinical severity scores from day 3 of treatment [74]. However, further trials must be conducted to reproduce these effects before this can be recommended as a therapy. Notably, chest physiotherapy and suctioning have not attained clinically meaningful endpoints in clinical trials either and are only recommended when there are underlying neuromuscular diagnoses that may impair an effective cough [48,49].

**Future Therapies and Directions:** High-dose inhaled nitric oxide (iNO) has also been suggested as another non-specific therapy for bronchiolitis due to its antibacterial, antiviral, and anti-inflammatory properties, as well as its bronchodilator effects. A small trial of 89 hospitalized infants with bronchiolitis examined four times daily and administered iNO treatment at 150 ppb for up to 5 days demonstrated a good tolerability and reduced time to clinical improvement when compared to the control [75]. Nonetheless, we acknowledge that there are currently no non-specific therapies that are recommended for use in acute RSV infection. This may be due to the fact that the term bronchiolitis describes a heterogeneous group of many distinct clinical entities and viruses that have been the inclusion criteria in clinical trials. Even in the case of RSV disease, it is widely recognized that there is a wide array of clinical phenotypes [66]. It may be that future trials will demonstrate efficacy if targeted therapies are matched to specific clinical phenotypes of RSV bronchiolitis, and this approach is currently being explored [76].

## 4. RSV Prevention

As the main therapeutic approach for acute RSV infection is supportive care, emphasis remains on the prevention of severe disease and hospitalization. Moreover, as sterilizing immunity to RSV is not achieved through infection and, thus, reinfection occurs throughout life in children and adults [77,78,79,80], the need for efficient, long-standing immunization is imperative. The first vaccination for RSV was assessed shortly after the first isolation of RSV in severely ill babies [81]. Unfortunately, the initial formalin-inactivated RSV vaccine showed that upon natural exposure to RSV, infants who were vaccinated experienced vaccine-enhanced RSV disease (ERD), with an 80% admission rate and the death of two infants [82]. The ERD phenomenon has since been studied extensively and is believed, in part, to be secondary to an exaggerated memory Th2 response, poor antibody affinity maturation, inadequate toll receptor signaling, and a low CD8 T-cell response [83,84]. ERD stalled the development of RSV vaccines for many years due to safety concerns; however, as our understanding of RSV structural biology and the mechanism of action has continued to evolve, there have been many advancements in RSV prevention strategies [85,86,87]. The different preventive approaches could be classified into two categories: firstly, passive immunization with monoclonal antibodies (mAB) or maternal vaccination during pregnancy, and secondly, active immunization by various types of vaccines designed for infants and adults (Figure 1). 

Active (A–D) and passive (E) immunization strategies for RSV are shown in Figure 1. Image created with Biorender.com.

### Vaccines

In the last decade, there has been a significant increase in the number of trials assessing different RSV vaccine strategies [85,86,88]. There are currently 34 different RSV vaccines in development, and 21 of these are currently advancing through Phase 1 to Phase 3 clinical trials [88]. 

RSV has several surface proteins, with two main glycoproteins as targets for immunization: the fusion (F) and attachment (G) proteins. Both are crucial for infectivity and viral pathogenesis [89] and, on the other hand, have a strong ability to induce protective neutralizing antibodies [90]. RSV has two distinct subtypes, RSV-A and RSV-B, which are distinguished mainly by variations in the G protein, while the F protein is more conserved [91,92], making the F protein a preferred target for the development of vaccines and mABs. Since the discovery that the F protein has two conformational forms, namely “pre-fusion F” (preF) and “post-fusion F” (postF), and the understanding that the preF conformation induces higher-potency neutralizing antibodies, it has become the preferred target for RSV-specific interventions [87,93,94].

The current RSV vaccine candidates can be divided into live-attenuated (LAV) or chimeric vaccines, as well as protein-based, recombinant-vector-based, and nucleic-acid-based vaccines [95]. 

**Recombinant-Vector-based Vaccines** use a modified replication-defective virus to induce a humoral and cellular immune response by delivering the genes of the relevant RSV proteins (antigens) [96,97]. In the recent SARS-COV-2 pandemic, adenovirus-based vaccines have demonstrated a good efficacy against severe disease [98] lending support to the argument that similar strategies for RSV may be successful, though concerns have arisen regarding an increased risk of thrombo-embolic events [99,100]. There are currently three recombinant-vector-based vaccine candidates in advanced clinical trials (phase 2–3), which are designed for the pediatric and the elderly populations. 

**Nucleic Acid Vaccines** mechanism is based on the introduction of messenger RNA (mRNA) encoding RSV antigens into the cells. In recent years, mRNA vaccines have shown safety and a high efficacy against SARS-CoV-2 infection [101,102,103], highlighting the potential of this approach. Although still in development, only one phase 1 trial of an RSV mRNA vaccine used in adults has been published [104], but with the knowledge obtained from the SARS-CoV-2 vaccine, it is likely that more trials will be held.

**Protein-based Vaccine** approaches (including whole-inactivated virus, particle, and subunit vaccines) are based on the display of various antigens with an increased density to create an enhanced immunologic reaction [105]. The current candidates are designed for older adults or young infants’ protection via maternal vaccination. Recently, an RSV F protein nanoparticle vaccination was assessed in pregnant women receiving vaccination between 28 and 36 weeks and did not meet the primary endpoint of a reduction in the rate of medically significant lower respiratory tract infections in the first 90 days of life. However, a potential benefit was found in regard to other outcomes, such as a decrease in severe infection with hypoxemia and decreased hospitalization [95]. There are other RSV protein-based vaccines in different stages of clinical trials (NCT04071158; NCT04785612; NCT04681833) [106,107,108,109] combining F and non-F antigens. 

**Live-attenuated vaccines** (LAV) mimic natural infection to generate a potent immune response while being attenuated for reduced virulence [110]. The main challenge with this class of vaccines is to achieve a favorable balance between safety (attenuation) and the creation of a strong immunogenic response [111]. LAVs do not appear to cause vaccine-enhanced disease in infants and are considered safe. A recent study assessing seven live-attenuated RSV vaccines administered intranasally to children aged 6–24 months demonstrated good efficacy rates. Compiled data of the five most promising vaccines showed an 88% efficacy rate against medically attended acute lower respiratory illness [110]. LAVs may provide important protection to older infants, who are not sufficiently protected by a mAb or maternal vaccine, as the effect of passive immunization wears off after a few months [105].

In a similar strategy, chimeric live virus vaccine candidates express RSV proteins in related attenuated viruses, such as Sendai and parainfluenza viruses. Chimeric vaccines have a good safety profile [112], although there are only a few chimeric RSV vaccine candidates currently in development.

**Monoclonal Antibodies:** The first strategy developed to provide passive immunization for the prevention of severe RSV infection was a mixture of human intravenous immunoglobulin (IVIG) containing high concentrations of RSV protective antibodies. In high-risk infants, this strategy was associated with a 40% reduction in RSV hospitalization, a 50% reduction in the number of hospitalization days, and a 60% reduction in the days of increased oxygen requirements [113,114]. However, adverse reactions (generally mild) occurred in 5% of the RSV IVIG infusions. 

The second strategy of RSV passive immunization was the development of palivizumab (Synagis®), a humanized mAB against the RSV fusion (F) glycoprotein, inhibiting RSV entry and infection [114]. Palivizumab is administered intramuscularly on a monthly basis during the RSV season. The selection of the population who receive this immune prophylaxis varies between jurisdictions, but the aim is to target high-risk infants (e.g., severe bronchopulmonary dysplasia, congenital heart disease, or severe immunodeficiency) in order to reduce severe disease in a cost-effective manner [115]. A recent Cochrane review found that prophylaxis with palivizumab reduced the rate of hospitalization due to RSV infection by 56% [116]. The main disadvantage of palivizumab is the cost and limited duration of effect based on the half-life of the antibody [19]. The need for repeated monthly administration is associated with missed doses, reducing the treatment’s overall efficacy [117], and the cost has limited its widespread use, especially in low-income countries [118]. As such, newer products with longer half-lives have been assessed in several clinical trials. 

Motavizumab is a second-generation mAB with a higher affinity for RSV. However, motavizumab did not demonstrate superiority over palivizumab in a phase 3 trial of high-risk children. Moreover, a trend of a higher rate of skin rashes was observed, and therefore, motavizumab was not approved by the FDA [119,120]. Suptavumab, a fully human monoclonal antibody targeting the prefusion F-protein-binding epitope, did not meet the trial efficacy endpoints in phase 3 due to its low efficacy against the predominant circulating RSV B strains in the trial [121]. 

Nirsevimab (Beyfortus^®^) is a recombinant human IgG1 kappa monoclonal antibody that binds the F1 and F2 subunits of the RSV fusion (F) protein at a highly conserved epitope [122]. This binding locks the RSV F protein in the prefusion conformation to block viral entry into the host cell. In a study assessing healthy preterm infants born at 29 to 34 weeks of gestational age, a single injection of nirsevimab administered before the RSV season resulted in a 70% reduction in the incidence of RSV-associated, medically attended LRTI (MALRTI) and a 78% reduction in the number of hospitalizations when compared to the placebo [123]. A more recent study assessing nirsevimab’s effect on healthy late-preterm and term infants showed a 75% reduction in MALRTI occurring up to 150 days after the injection in the nirsevimab group versus placebo, without a change in the admission rates [124]. These differences were consistent throughout the study period and across RSV subtypes. Importantly, the safety profile of nirsevimab was reported to be similar to that of palivizumab [125]. Recently, nirsevimab was approved in the European Union for the prevention of RSV disease in newborns and infants during their first RSV season [126].

**Figure 1 pathogens-12-00154-f001:**
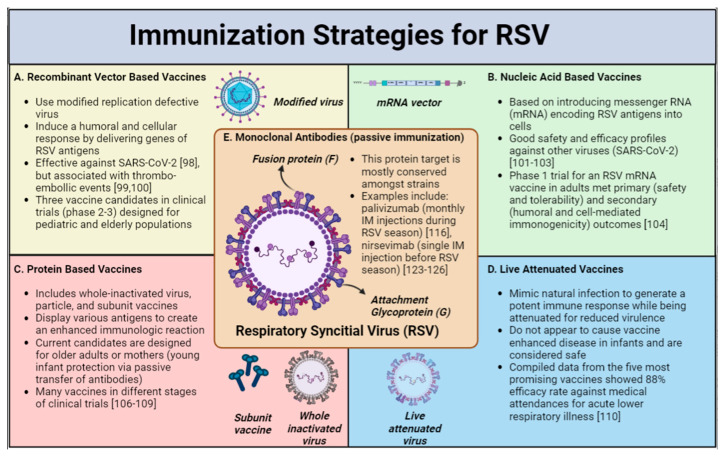
Immunization Strategies for RSV [98,99,100,101,102,103,104,106,107,108,109,110,116,123,124,125,126].

Further studies studying the efficacy of these mABs will help to guide prophylaxis efforts, as a long-half-life product that eliminates the need for repeated administrations is very appealing. In the interim, there are ongoing efforts to render mAB products more affordable [118,127] and easier to administer with the use of needle-free administration [128]. 

To summarize, there are currently many maternal and infant RSV vaccines in different clinical trials, together with the continuous evaluation of newer mABs, with some of these studies showing promising results. The future management of RSV prevention will likely combine the different strategies of active and passive immunization. In the neonatal and early infancy periods, the immunization goals may be achieved by passive mAB administration or maternal vaccinations, or a combination of both. As passively acquired immune responses wane over time, active immunization through vaccines will provide complementary protection for the older pediatric and adult age groups.

## 5. Expert Commentary

Despite tremendous progress in our understanding of respiratory viral infections, these pathogens remain a significant cause of morbidity and mortality around the world. In general, there are limited specific therapies for any respiratory virus, and the mainstay of management focuses on prevention. This paradigm is also reflected in RSV, and while the lack of an efficacious RSV-specific therapy remains a concern, the current situation leads us to make three specific comments.

First, prevention is better than treatment. From this review, we can see that the existing measures used to prevent RSV are efficacious, and future strategies (vaccines, long-acting mABs) also offer the potential for further gains. The COVID-19 pandemic has highlighted, on a global scale, that prevention can be immensely effective. Specifically, the social measures that were enforced to reduce the spread of SARS-CoV-2 also impacted other respiratory viruses, including RSV. Thus, in a Canadian study, for example, RSV was essentially eliminated in the 2020–2021 season [129]. However, this form of prevention has led to a population with little or no immunity to these viruses, and the 2022–2023 viral season will likely demonstrate high rates of infection if social isolation measures are reduced. Thus, the prevention of disease needs to be associated with protection from disease (i.e., immunity) so as to ensure protection in the future. Vaccines offer the best option for improving the morbidity and mortality associated with RSV.

Second, the lack of an RSV treatment after much effort raises questions as to the feasibility of treating RSV. It is conceivable that the clinical presentation of an RSV infection in a typical setting renders it exceedingly difficult to treat. To illustrate this point, consider that in 2014, after many years of pre-clinical work, GS-5806, an RSV fusion inhibitor, was tested in a human RSV challenge model and was shown to be an effective RSV treatment [130]. This led to further investment and a number of clinical trials aiming to study the real-world potential of GS-5806. In 2020, a review summarized the findings of four trials of presatovir (GS-5806) and found that the treatment did not meet the study endpoints in any of the trials [131]. Thus, while in controlled settings this novel therapy worked, in a clinical situation, there was no demonstrated benefit. In the real world, by the time people present with symptomatic RSV or other respiratory infections, the infection of the epithelium has often progressed to such a degree that the inhibition of further viral infection has a limited impact. Thus, to increase the potential of a future therapeutic to be beneficial, novel strategies will be required so as to identify infection earlier, or alternatively, we will need to consider a post-exposure prophylaxis model in high-risk individuals. The current strategies used to treat symptomatic patients that present when unwell will handicap existing and future therapeutics. 

Finally, from this review, it is clear that while specific therapies are lacking and the current prevention strategies offer great potential, the best treatment for RSV infection remains supportive care. This does not suggest that supportive care is not beneficial in the setting of RSV. The mortality gap for RSV between low- and middle-income countries in comparison to the developed world, in part, highlights that the existing approaches can reduce death. RSV results in more than 100,000 infant deaths around the world, but ~97% of these deaths are in low- and middle-income countries [5]. In order to maximize the impacts on RSV morbidity and mortality, rather than investing in the development of novel (and often expensive) therapies, resources should be better and more ethically spent on improving the current situation in low- and middle-income countries.

## Figures and Tables

**Table 1 pathogens-12-00154-t001:** Non-specific approaches to treating acute RSV bronchiolitis.

Product Name	Mode of Administration/Mechanism of Action	Recommendation
*Mucus therapies*		
Deoxyribonuclease (hrDNase)	Nebulized solution Mucolytic compound Cleaves extracellular DNA	Not recommended [35] May be considered as a therapeutic option for atelectasis in severe cases, when conventional therapy is unsuccessful [36]
N-acetylcysteine	Nebulized solution Poor bioavailability of oral preparation [37] Mucolytic compound Hydrolyzes disulfide bonds of mucus proteins Antioxidant properties [38]	Insufficient data—further studies needed Not recommended
3% hypertonic saline	Nebulized solution Creates osmotic gradient and pulls water into the mucus layer Improves ciliary activity Stimulates cough May reduce airway edema [39]	May reduce the risk of hospitalization in the ED setting [40] Not recommended for inpatient management May modestly reduce the duration of admission for infants admitted >72 h—further studies needed [40,41]
*Bronchodilators*		
Salbutamol, albuterol, etc.	Nebulized solution β-2 adrenergic receptor agonist, relaxes smooth muscle and opens airways	Not recommended [33] Can be trialled to assess the response in certain cases and given where a benefit is seen
Epinephrine	Nebulized solution Some β-2 adrenergic effects Vasoconstriction due to α-1 adrenergic receptor effects also decrease airway edema	May reduce risk of hospitalization in the ED setting [42] Not recommended [43]
*Therapies targeting inflammation*		
Glucocorticoids (dexamethasone, prednisolone, budesonide, etc.)	Oral solution, inhaled or nebulized preparations Broad spectrum anti-inflammatory Repress the expression of pro-inflammatory cytokines	Not recommended [44,45] Could be considered where reactive airway disease is strongly suspected (asthma, bronchopulmonary dysplasia, etc.), but no good evidence
Leukotriene inhibitors (montelukast, etc.)	Oral solution Inhibit leukotrienes, which are endogenous mediators of inflammation [46]	Poor evidence, not recommended [47]
*Manual therapies*		
Chest physiotherapy	Chest percussion, suction Aids in the clearance of secretions Thought to decrease ventilatory effort for infants on the severe end of the disease spectrum	Not routinely recommended [48] Can be considered when relevant comorbidities are present (neuromuscular conditions, etc.) [49]

## Data Availability

Not applicable.

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
