# Peer review of "Prevention and Treatment Strategies for Respiratory Syncytial Virus (RSV)"

_pathogens, 2023, doi:10.3390/pathogens12020154_

Round 1

Reviewer 1 Report

This is an excellent up to date review and a pleasure to read.

Author Response

This is an excellent up to date review and a pleasure to read.
We thank the reviewer for their review and comment.

Reviewer 2 Report

This is a well written and comprehensive review that covers an extensive overview of RSV specific therapy and non-specific therapy, with a strong focus on clinical benefit but also in context of RSV pathophysiology. It further describes immunization strategies for RSV, highlighting different types of vaccines and monoclonal antibodies, their clinical stage and benefits. In the expert commentary the review emphasizes that there are still limited specific therapy for any respiratory virus, including RSV and the need for early administration of therapy. Overall, the manuscript is well organized and well written.

I have noticed a few very minor revisions that would aid in clarity:

p.3 l. 113 please include ‘amongst other compounds’

p.4 Table 1: N-acetylcysteine: I would recommend to add that the outcome is not clear and further studies are needed

p. 3 l. 115 For treatments that are available on the market, the brand name needs to be included, such as Palivizumab (Synagis®).

It would be beneficial to include clinical trials numbers for the most advancing vaccines and therapies in the main body of the text and not only in the references.

p. 12 References: Different style of references were noted, please revise them carefully (for example references 61 and 97 are capitals only and different to other references)

Overall, I found this an interesting review that will be useful for those working on RSV and I thank the authors for putting it together.

Reviewer 3 Report

Comments:

In the review, authors have summarized the status of current treatment regimens including therapeutic, passive (antibody based) and active (vaccine based) apocopates to control Respiratory Syncytial 1 Virus (RSV) infection and strategies to enhance protective immune response. Authors have also highlighted current vaccine efforts for the development of better vaccines/therapeutics and antigens of interest for each vaccine type.

Most effective vaccines combined the contribution for cellular and humoral responses. Current vaccines efforts to many viruses (for example influenza) including RSV, are focused on generation of CD4 T cells mediated protection. I would like for authors to site, apart from mRNA-based RSV prefusion F protein vaccine (Ref 104),

if any of vaccine efforts are focused on seeding CD4 memory pools to help virus specific B cells to boost neutralizing antibody responses.

Overall, review covers most of the aspects on RSV immunity, our understating and where we stand for development of better RSV vaccine.

Author Response

Reviewer 3
In the review, authors have summarized the status of current treatment regimens including therapeutic, passive (antibody based) and active (vaccine based) apocopates to control Respiratory Syncytial 1 Virus (RSV) infection and strategies to enhance protective immune response. Authors have also highlighted current vaccine efforts for the development of better vaccines/therapeutics and antigens of interest for each vaccine type.
Most effective vaccines combined the contribution for cellular and humoral responses. Current vaccines efforts to many viruses (for example influenza) including RSV, are focused on generation of CD4 T cells mediated protection. I would like for authors to site, apart from mRNA-based RSV prefusion F protein vaccine (Ref 104),
if any of vaccine efforts are focused on seeding CD4 memory pools to help virus specific B cells to boost neutralizing antibody responses.
Overall, review covers most of the aspects on RSV immunity, our understating and where we stand for development of better RSV vaccine.

We thank the reviewer for raising this perspective. We are unaware of any specific vaccine strategy that directly targets the memory CD4 pool. Undoubtably, some approaches may lead to an expansion of various memory subsets but we could not find a reference that specifically focused on enhancing CD4 memory cells against RSV. If the reviewer has knowledge of a specific paper (or papers) that they feel would strength out current review, we would be pleased to include it (or them).